:ᐭ: PLOS | ONE

# Hepatopulmonary syndrome has low prevalence of pulmonary vascular abnormalities on chest computed tomography

Luciano Folador[1,2]*, Felipe S. Torres[3], Juliana F. Zampieri[2], Betina C. Machado[1], Marli M. Knorst[1,4], Marcelo B. Gazzana[1,4]

1 Programa de Pós-Graduação em Ciências Pneumológicas, Faculdade de Medicina, Universidade Federal do Rio Grande do Sul, Porto Alegre, RS, Brazil, 2 Serviço de Radiologia, Hospital de Clínicas de Porto Alegre, Porto Alegre, RS, Brazil, 3 Department of Medical Imaging, University of Toronto, Toronto, ON, Canada, 4 Serviço de Pneumologia, Hospital de Clínicas de Porto Alegre, Porto Alegre, RS, Brazil

* lucianofolador@gmail.com

**Data Availability Statement:** All relevant data are within the manuscript and its Supporting Information files.

## Abstract

### Purpose

Hepatopulmonary syndrome (HPS) is defined as an arterial oxygenation defect induced by intrapulmonary vascular dilatations associated with hepatic disease. This study aimed to assess the prevalence of type 1 and 2 pulmonary vascular abnormalities on chest computed tomography (CT) in patients with cirrhosis and HPS and to characterize intra- and interobserver reliability.

### Materials and methods

Two thoracic radiologists retrospectively evaluated chest CT scans from 38 cirrhosis patients with HPS. They classified the pulmonary vascular abnormalities as type 1 (multiple dilated distal pulmonary arteries), type 2 (nodular dilatation or individual pulmonary arterial malformation), or absence of abnormality. Furthermore, they measured the diameters of the central pulmonary arteries and subsegmental pulmonary arteries and bronchi. We analyzed the prevalence, intraobserver reliability, and interobserver reliability of abnormal CT findings related to HPS, and the correlation of these findings with partial arterial oxygen pressure ($PaO_2$).

### Results

The overall prevalence of pulmonary vascular abnormalities was 28.9% (95% confidence intervals: 15.4%, 45.9%). Moreover, 26.3% of patients had type 1 abnormality (13.4%, 43.1%) and 2.6% of patients had type 2 abnormality (0.0%, 13.8%). The intraobserver reliability kappa value was 0.666 (0.40, 0.91) and the interobserver kappa value was 0.443 (0.12, 0.77). There was no correlation between pulmonary vascular abnormalities on CT and $PaO_2$ values.

**Funding:** The author(s) received no specific funding for this work.

**Competing interests:** The authors have declared that no competing interests exist.

**Abbreviations:** $AaO^2$, alveolar-arterial oxygen gradient; ABR, artery-to-bronchus ratio; CT, computed tomography; HPS, hepatopulmonary syndrome; ICC, intraclass correlation coefficients; MDCT, Multidetector row computed tomography; $PaO_2$, partial pressure of oxygen; 95%CI, 95% confidence interval.

## Conclusions

The prevalence of pulmonary vascular abnormalities on chest CT of patients with cirrhosis and HPS is low and not correlated with $PaO_2$. These findings question the usefulness of chest CT for the evaluation of patients with cirrhosis and HPS.

## Introduction

HPS affects 3–47% of patients with terminal liver disease depending on the diagnostic criteria [1–5], and it is an independent risk factor for a worse prognosis among cirrhosis patients [5,6].

There is little, and contradictory, information regarding the use of thoracic computed tomography (CT) scans for the diagnosis of HPS [1]. Initial data from a small study with only 10 patients reported that multiple dilated vessels with increased numbers of visible terminal branches that extend to the pleura indicate a diagnosis of HPS [7]. Two subsequent studies in individuals with cirrhosis and HPS reported an increased CT diameter of the peripheral pulmonary vasculature compared with both healthy controls and patients with normoxemic cirrhosis [8,9]. In contrast, another study reported evidence of peripheral pulmonary artery dilatation on CT in patients with liver disease compared with healthy controls, but no differences between patients with liver disease with or without HPS [10].

In addition, some authors [11,12] have extrapolated to CT data from a pulmonary angiographic study of just 7 patients [13], which classified patterns of pulmonary vascular abnormalities as type 1 (distal vascular dilatation with multiple vessels extending toward the pleura and subpleural space) and type 2 (arteriovenous malformations and nodular dilatations of the peripheral vessels). However, no previous studies have addressed the prevalence of these two patterns of pulmonary vascular abnormalities in a larger sample of patients with HPS or evaluated the reproducibility of CT pulmonary vasculature and bronchial-to-artery ratio measurements in such patients.

Therefore, the purpose of this study was to assess the prevalence of type 1 and 2 pulmonary vascular abnormalities on chest CT in patients with cirrhosis and HPS. We also aimed to characterize intraobserver and interobserver reliability of qualitative and quantitative abnormal CT findings related to HPS. Finally, we analyzed correlations between these findings and partial arterial oxygen pressure ($PaO_2$), a marker of disease severity.

## Materials and methods

### Patients

Data from consecutive patients of the pulmonary circulation clinic at a university tertiary care hospital from January 2010 to December 2016 were reviewed. Patients were included where they had a chest CT and a diagnosis of HPS based on the following criteria: (i) presence of chronic liver disease; (ii) alveolar–arterial oxygen gradient ($AaO_2$) >15 mmHg (20 mmHg in patients over 64 years old) detected with blood gas analysis; and (iii) demonstration of intrapulmonary vascular dilatation by means of a positive contrast-enhanced echocardiography or perfusion lung scanning with technetium-99m-labelled macroaggregated albumin [1,2].

Results from contrast-enhanced echocardiography, clinical exams, blood tests, and arterial blood gas in room air were recorded from electronic patient records. Patients with a $PaO_2$ in room air below 80 mmHg were considered hypoxemic and above this as normoxemic [1]. The maximum interval tolerated between chest CT and blood gas analysis was three months.

Participants with pulmonary hypertension, bronchiectasis, interstitial lung disease, severe chronic obstructive pulmonary disease, moderate/large pleural effusion, and/or those with severe motion artefacts on chest CT were excluded.

The local institutional review board approved the study protocol. Due to its retrospective nature the board waived the need for written informed consent. There are no conflicts of interest to declare.

## CT Acquisition and analysis

Images were obtained with a 8-multidetector row CT (MDCT) scanner (BrightSpeed Edge, GE Medical Systems, USA), 16-MDCT scanner (Brilliance 16, Philips Healthcare, the Netherlands) or 64-MDCT scanner (Aquilion 64, Toshiba Medical Systems, Japan), with patients in the supine position and at full inspiration. All scans were volumetric acquisitions (slice thickness: 1.0–2.0 mm) and were reconstructed with a high spatial frequency algorithm. Images were stored and analyzed with a picture and archiving communication system (IMPAX 6.6.1.3525, Aghfa HealthCare, Belgium) and all measurements were conducted by manually placing an electronic caliper tool. When used, iodinated nonionic intravenous contrast media was injected in a peripheral vein at a dose of 1–2 ml/kg of body weight.

Two thoracic radiologists (Observer 1 [LF], 5 years of experience, and observer 2 [JFZ], 3 years of experience) independently evaluated all the images. The diameters of the main pulmonary artery, right and left pulmonary arteries, and subsegmental pulmonary arteries and bronchi were measured. When present, pulmonary vascular abnormalities were classified as type 1 or type 2. In case of discrepancies between the two assessors regarding the presence and type of pulmonary vascular abnormality, a third independent thoracic radiologist (FST, 10 years of experience) evaluated the images. Observer 1 performed all the measurements twice, with an interval of six months to a year to prevent recall bias. Interobserver reliability was calculated based on his first evaluation. Intraobserver reliability was calculated only for the observer 1. Assessors were blinded to the clinical and laboratory parameters of the patients, as well as to all measurements for the other assessors.

The presence of pulmonary vascular abnormalities, defined as Type 1 or Type 2, was evaluated at lung window (W: 1500 HU, L: -500 HU). Type 1 abnormality (Fig 1) was defined as the presence of dilated peripheral arteries that do not taper normally and that touch the pleural surface in the lower lobes. Type 2 abnormality (Fig 2) was characterized by the presence of individual arteriovenous malformations and nodular dilatation of peripheral pulmonary vessels [12].

Quantitative assessment of peripheral arteries was performed by calculating the artery-to-bronchus ratio (ABR). The ABR was defined as the ratio between the diameter of a 4th to 6th generation subsegmental artery and the outer diameter of its accompanying bronchi in each lower lobe. On lung window (W: 1500 HU, L: -600 HU), observers independently chose two different subsegmental arteries and its accompanying bronchi in each lower lobe on a axial image where both were rounded and parallel to each other. A single ABR value per patient was calculated from the average of measurements performed in each lower lobe. Central pulmonary arteries were measured at mediastinal window (W: 350 HU, L: 30 HU). The diameter of the main pulmonary artery was measured at the level of its bifurcation, perpendicular to its long axis. The diameters of the right and left pulmonary arteries were measured at their widest portion before branching.

## Statistical analysis

Statistical analysis was performed with SPSS version 20 (IBM, USA). Categorical variables are presented as frequencies and percentages. Prevalence is presented with its respective 95%

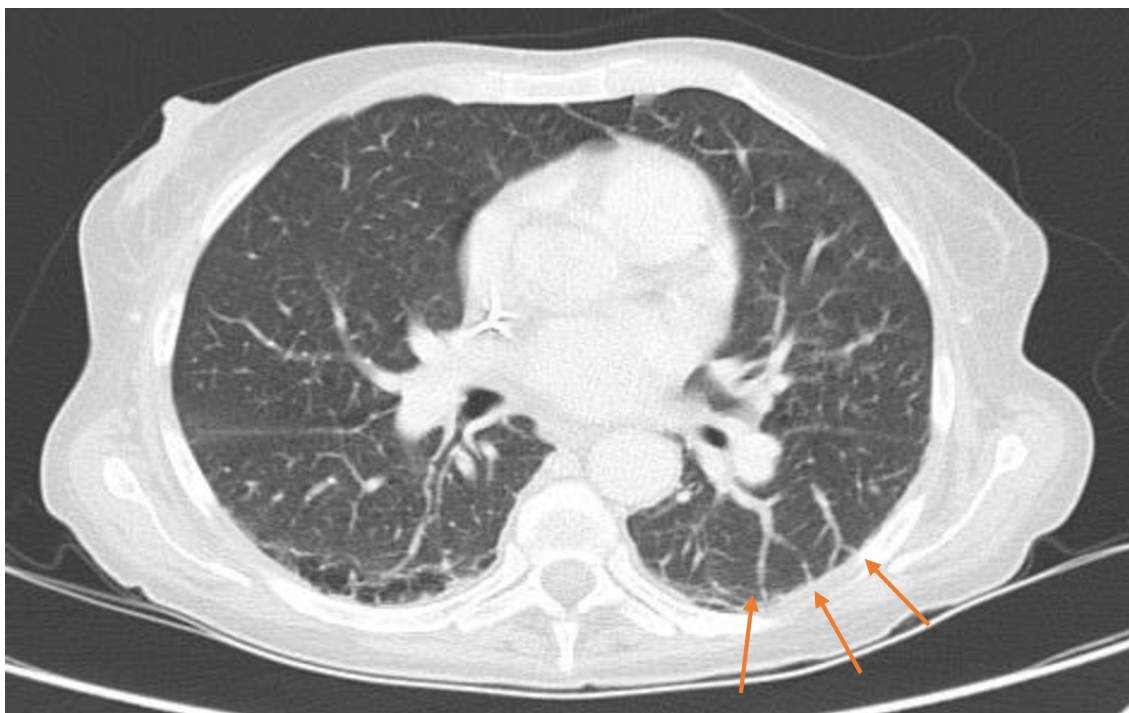

**Fig 1. Type 1 pulmonary vascular abnormality.** Type 1 pulmonary vascular abnormality was defined as the presence of dilated peripheral arteries that do not taper normally and that touch the pleural surface in the lower lobes (arrows).

confidence interval (95%CI). Quantitative variables were evaluated for their symmetry with the Kolmogorov Smirnov test and are presented as mean and standard deviation. Categorical variables were evaluated with Chi-square test or Fisher's exact test. Quantitative variables were compared by using Student's t-test for independent samples. A p-value of $<0.05$ was considered statistically significant.

To assess the concordance between categorical variables, the Kappa coefficient (K) of agreement was used. Bland and Altman plots, and intraclass correlation coefficients (ICC) were used to assess reliability among quantitative variables. Kappa and ICC values of 0.00–0.20 were considered as slight agreement; 0.21–0.40 as fair agreement; 0.41–0.60 as moderate agreement; 0.61–0.80 as substantial agreement, and 0.81–1.00 as almost perfect agreement [14]. In addition, 95% CIs for each value of K and ICC were calculated.

## Results

During the study period, 53 patients met the inclusion criteria. However, 15 patients were excluded for the following reasons: large pleural effusion (n = 3); severe respiratory motion artefacts on CT (n = 2); and an interval between chest CT and blood gas analysis greater than three months (n = 10). The characteristics of the 38 patients included in the study are summarized in Table 1.

The overall prevalence of either type 1 or type 2 pulmonary vascular abnormalities was 28.9% (95% CI: 15.4%, 45.9%), with 26.3% (13.4%, 43.1%) being Type 1 and 2.6% (0.1%, 13.8%) being Type 2 (Table 2). In the first analysis, Observer 1 identified Type 1 pulmonary vascular abnormalities on CT in 10/38 patients (26.3%), Type 2 pulmonary vascular abnormality in 1/38 patient (2.6%), and no pulmonary vascular abnormality in 27/38 patients (71.1%). In the second analysis, Type 1 abnormality was detected in 14/38 patients (36.8%), Type 2 in 1/

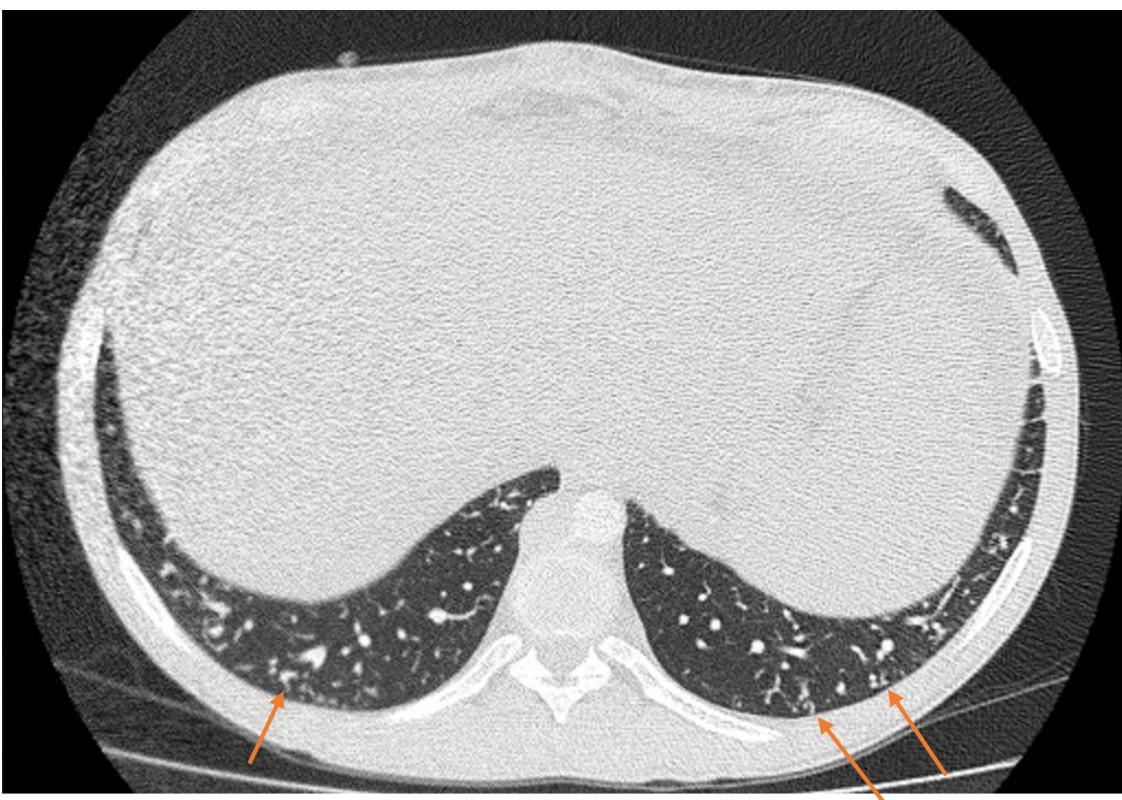

**Fig 2. Type 2 pulmonary vascular.** Type 2 pulmonary vascular was characterized by the presence of individual arteriovenous malformations and nodular dilatation of peripheral pulmonary vessels (arrows).

38 patient (2.6%), and no pulmonary vascular abnormality in 23/38 patients (60.5%). Observer 2 identified Type 1, Type 2, and no abnormality in 6/38 (15.8%), 1/38 (2.6%), and 31/38 patients (81.6%), respectively. The intraobserver Kappa value was 0.67 (95% CI: 0.40, 0.91; p<0.001) and the interobserver Kappa value was 0.44 (95% CI: 0.12, 0.77; p = 0.02) (Table 3).

There was no difference in the prevalence of pulmonary vascular abnormalities when comparing normoxemic (n = 20) and hypoxemic patients (n = 18) (p = 0.513). The mean $PaO_2$ of patients with any vascular pulmonary abnormality on CT (n = 11) was 82.1 mmHg ± 17.9 and for those without any pulmonary vascular abnormality (n = 27) it was 77.9 mmHg ± 15.8 (p = 0.515). There was no correlation between $PaO_2$ and ABR (Fig 3).

Table 4 presents the measurements of the peripheral and central pulmonary vasculature. Intra- and interobserver limits of agreement with the Bland and Altmann technique in measurements of ABR, main pulmonary artery, right pulmonary artery, and left pulmonary artery are shown in Figs 4 and 5. Values for intra- and interobserver reliability are presented in Table 3.

## Discussion

Here, we demonstrate a low prevalence of pulmonary vascular abnormalities on chest CT in patients with cirrhosis and HPS. Moreover, while this qualitative variable has good intraobserver agreement, interobserver agreement was only moderate. Furthermore, intra- and interobserver reliability and correlation with the Bland and Altmann technique of caliper-based measurements of peripheral pulmonary vessels and ABR were poor. There was no correlation

**Table 1. Patient demographics and clinical characteristics.**

| Variable | Value |
|---|---|
| Male sex (%) | 20 (52.6%) |
| Age (years) | 54.11 ± 10.02 |
| Liver disease etiology (%) | |
| Hepatitis C | 26 (68.4%) |
| Cryptogenic | 2 (5.3%) |
| Alcoholic | 4 (10.5%) |
| NASH | 2 (5.3%) |
| Hepatitis B | 1 (2.6%) |
| Child-Pugh class distribution | |
| A | 14 (36.8%) |
| B | 16 (42.1%) |
| C | 8 (21.1%) |
| MELD score | 13.22 ± 4.70 |
| Smoking (%) | |
| Never | 18 (47.4%) |
| Current | 9 (23.7%) |
| Former | 11 (28.8%) |
| Pulmonary disease (%) | |
| None | 32 (84.2%) |
| Mild COPD | 6 (15.8%) |
| Heart disease (%) | |
| None | 35 (92.1%) |
| Ischemic | 3 (7.9%) |
| $PaO_2$ (mmHg) in room air | 79.16 ± 16.30 |
| $AaO_2$ (mmHg) in room air | 31.18 ± 14.32 |
| Severity grade of HPS (%) | |
| Mild ($PaO_2 \geq 80$ mmHg) | 20 (52.6%) |
| Moderate (80 mmHg< $PaO_2 \geq 60$ mmHg) | 12 (31.6%) |
| Severe (60 mmHg< $PaO_2 \geq 50$ mmHg) | 2 (5.3%) |
| Very severe ($PaO_2 < 50$ mmHg) | 4 (10.5%) |
| $PaO_2$ (mmHg) in 100% $O_2$ | 400.65 ± 110.47 |
| Shunt (%) in 100% $O_2$ | 15.62 ± 5.39 |
| IPVD (shunt) grade by echocardiography (%) | |
| I | 21 (55.3%) |
| II | 7 (18.4%) |
| III | 3 (7.9%) |
| IV | 7 (18.4%) |
| Intravenous contrast on CT (%) | |
| Yes | 15 (39.5%) |
| No | 23 (60.5%) |
| Interval between CT and blood gas analysis (days) | 35.0 ± 29.5 |

Note.–Mean values are provided with standard deviations. NASH = non-alcoholic steatohepatitis. MELD = Model for End-Stage Liver Disease. COPD = chronic obstructive pulmonary disease. $PaO_2$ = partial pressure of oxygen. $AaO_2$ = alveolar–arterial oxygen gradient. IPVD = Intrapulmonary vascular dilatation. CT = computed tomography

between abnormal CT findings and $PaO_2$. This finding questions the usefulness of pulmonary vascular assessment using thoracic CT in patients with cirrhosis and HPS.

**Table 2. Prevalence of peripheral pulmonary vascular abnormalities.**

|  | Type 1 | Type 2 |
|---|---|---|
| Prevalence | 26.3% (13.4%, 43.1%) | 2.6% (0.07%, 13.81%) |
| n | 10/38 | 1/38 |

**Table 3. Intraobserver and interobserver reliability of qualitative and quantitative abnormal computed tomography findings.**

|  | Intraobserver | Interobserver |
|---|---|---|
| Pulmonary vascular abnormality | K = 0.666 (0.40, 0.91) | K = 0.443 (0.12, 0.77) |
| Artery-to-bronchus ratio | ICC = 0.553 (0.287, 0.740) | ICC = 0.368 (0.059, 0.613) |
| Main pulmonary artery | ICC = 0.786 (0.626, 0.883) | ICC = 0.837 (0.708, 0.912) |
| Right pulmonary artery | ICC = 0.840 (0.713, 0.913) | ICC = 0.827 (0.692, 0.906) |
| Left pulmonary artery | ICC = 0.794 (0.638, 0.887) | ICC = 0.844 (0.719, 0.916) |

Note.–K = kappa coefficient. ICC = intraclass coefficient

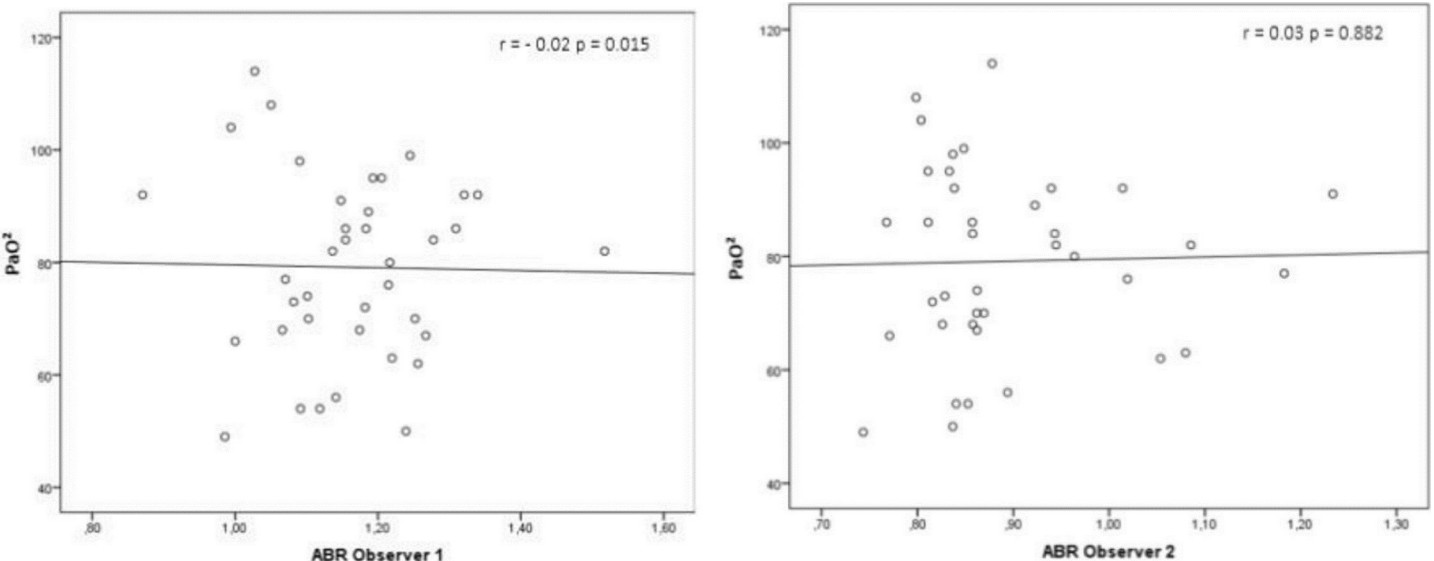

**Fig 3. Linear correlation between PaO$_2$ and ABR.** Linear correlation between partial pressure of oxygen (PaO$_2$) and the artery-to-bronchus ratio (ABR) for observer 1 and 2.

**Table 4. Measurements of peripheral and central pulmonary vasculature.**

|  | Observer 1: first measurement | Observer 1: second measurement | Intraobserver comparison | Observer 2 | Interobserver comparison |
|---|---|---|---|---|---|
| Artery-to bronchus ratio | 1.16 ± 0.11 | 1.12 ± 0.13 | p = 0.03 | 0.89 ± 0.11 | p = 0.01 |
| Main pulmonary artery (mm) | 25.56 ± 2.52 | 25.55± 2.63 | p = 0.90 | 24.98 ± 2.48 | p = 0.01 |
| Right pulmonary artery (mm) | 20.98 ± 2.30 | 21.42 ± 2.55 | p = 0.059 | 20.51 ± 2.30 | p = 0.03 |
| Left pulmonary artery (mm) | 20.42 ± 2.05 | 21.16 ± 2.21 | p = 0.02 | 19.77 ± 1.98 | p = 0.01 |

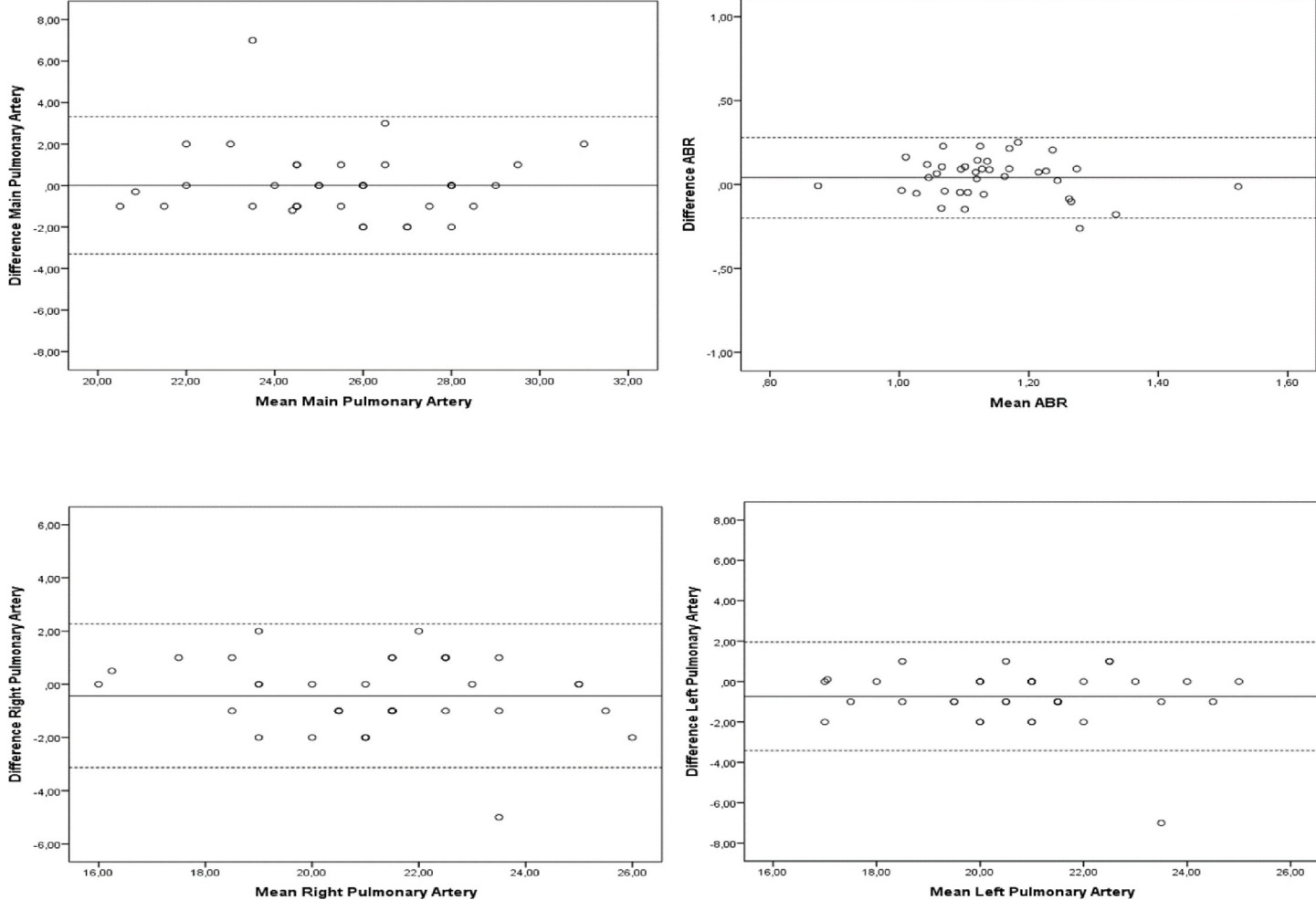

**Fig 4. Intraobserver limits of agreement.** Intraobserver limits of agreement with the Bland and Altmann technique.

No previous study has reported the prevalence of pulmonary vascular abnormalities on chest CT and evaluated intra- and interobserver reliability of this qualitative finding. However, one study reported prevalence of Type 1 (85%) and Type 2 pulmonary vascular abnormalities (15%) in cirrhosis patients with HPS [13]. Interestingly, several authors have cited and referenced these data, extrapolating the values to CT imaging [11, 12]. However, this prevalence was established with conventional pulmonary angiography among just 7 patients and performed by one observer [13]. Here, in a larger sample of 38 patients with cirrhosis and HPS, we demonstrate a lower prevalence of pulmonary vascular abnormalities with chest CT.

The lower prevalence of pulmonary vascular abnormalities in our study may be due to the spatial resolution of CT. Intrapulmonary vascular dilatation is characterized by dilatations of alveolar septal arterioles and capillaries with a normal diameter of 7–15 μm to 15–150 μm [1, 15]. This is equivalent to the inferior limit of the minimum spatial resolution of CT (100–300 μm) [16]. The lack of correlation between pulmonary vascular abnormalities and $PaO_2$, a marker of disease severity in HPS patients [1], may be because CT underestimates the presence of dilatation of microscopic peripheral vessels.

The mean ABRs in our and other studies [7–10] are within the normal range reported for healthy patients (0.98; 95% CI: 0.7, 1.26) [17]. Moreover, consistent with previous reports [10],

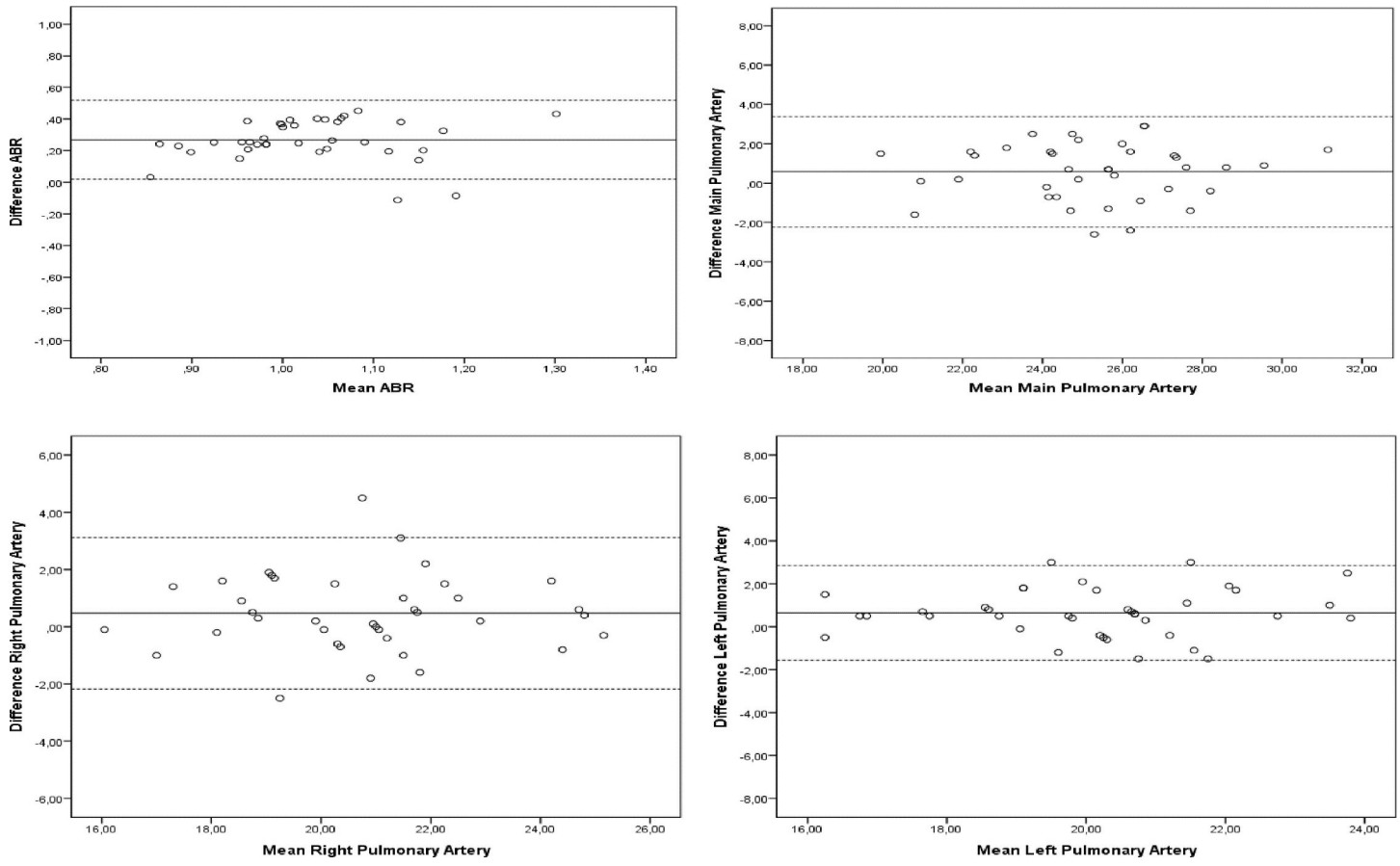

**Fig 5. Interobserver limits of agreement.** Interobserver limits of agreement with the Bland and Altmann technique.

ABR and PaO$_2$ were not correlated. Chen et al. [10] compared patients with HPS and those with liver dysfunction without HPS, reporting no difference in ABR. Our study reinforces the hypothesis that dilatation of muscularized pulmonary arterioles is a consequence of the hyperdynamic circulatory state of cirrhosis mediated by vasoactive substances. Intrapulmonary vascular dilatations constitute a distinct anatomic abnormality located distal to these vessels [10].

Two previous studies [8, 9] demonstrated higher ABRs in patients with HPS compared with those with cirrhosis without HPS. Koksal et al. [8] reported a weak negative correlation (r = -0.64; P = 0.04) between the diameters of right lower lobe basal segmental pulmonary arteries and PaO$_2$ in patients with HPS. This correlation may be higher than ours because one of the 10 patients had a basilar arterial diameter of 9 mm, which more likely represents a pulmonary arteriovenous malformation [10]. Lee et al. [9] also demonstrated dilatation in peripheral pulmonary vasculature in patients with cirrhosis and HPS. However, the presence of intrapulmonary vascular dilatation was not confirmed with contrasted-enhanced echocardiography or perfusion lung scanning with technetium-99m-labelled macroaggregated albumin.

Our study also illustrates the challenge of measuring small structures, such as subsegmental arteries and bronchi, with CT in clinical practice. The diameters of central pulmonary vessels had almost perfect intra- and interobserver agreement, while ABR had weak intra- and interobserver agreement. This is expected since reliability of measurements decreases as the size of structures decrease [18]. In fact, manual tracing of inner and outer contours of the airway cross-section on axial CT images suffers from large intra- and interobserver variability in

airway measurement [19]. This may explain why we observed a significant difference in ABR between observers. Automatic or semi-automatic ABR measurement could be an option. However, these systems show large variation and fail to correctly pair arteries and airways in 21.9% of cases [20]. Given the weak interobserver reliability of ABR measurements in our and other studies [19], the use of this technique to evaluate peripheral vascular dilatation is questionable and affects the external validity of previous papers.

Our study has several limitations, including its retrospective nature and the absence of a specific thin slice high resolution CT protocol. Nonetheless, the slice thickness and the reconstruction algorithms used are the ones currently selected in clinical practice. Additionally, the 3-month interval between CT and blood gas analysis may have affected the correlation between $PaO_2$ and the CT findings, particularly in borderline normoxemic or hypoxemic patients.

In conclusion, this is the first study to access the prevalence of type 1 and 2 pulmonary vascular abnormalities on chest CT in patients with cirrhosis and HPS (26.3% and 2.6%, respectively). The prevalence of these abnormalities on Chest CT were not correlated with $PaO_2$. The intra- and interobserver reliabilities of this qualitative finding were only moderate. Our study suggests that CT has a limited role in the diagnosis of HPS and its main utility remains in excluding other underlying pulmonary diseases.

## Supporting information

**S1 Table. Data before analysis.**
(XLS)

## Author Contributions

**Conceptualization:** Luciano Folador, Felipe S. Torres, Juliana F. Zampieri, Betina C. Machado, Marli M. Knorst, Marcelo B. Gazzana.

**Data curation:** Luciano Folador, Felipe S. Torres, Juliana F. Zampieri, Betina C. Machado.

**Formal analysis:** Luciano Folador.

**Investigation:** Luciano Folador, Felipe S. Torres, Juliana F. Zampieri, Betina C. Machado.

**Methodology:** Luciano Folador, Felipe S. Torres, Marli M. Knorst, Marcelo B. Gazzana.

**Project administration:** Luciano Folador, Marcelo B. Gazzana.

**Supervision:** Felipe S. Torres, Marli M. Knorst, Marcelo B. Gazzana.

**Visualization:** Felipe S. Torres, Marcelo B. Gazzana.

**Writing – original draft:** Luciano Folador, Felipe S. Torres, Marcelo B. Gazzana.

**Writing – review & editing:** Luciano Folador, Felipe S. Torres, Juliana F. Zampieri, Betina C. Machado, Marli M. Knorst, Marcelo B. Gazzana.

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
