## [Decision Letter · Decision Letter 0]

6 Sep 2019

[EXSCINDED]

PONE-D-19-21631

Prevalence of pulmonary vascular abnormalities on chest computed tomography related to hepatopulmonary syndrome and relationship with disease severity

PLOS ONE

Dear Dr. Folador,

Thank you for submitting your manuscript to PLOS ONE. After careful consideration, we feel that it has merit but does not fully meet PLOS ONE’s publication criteria as it currently stands. Therefore, we invite you to submit a revised version of the manuscript that addresses the points raised during the review process.

The manuscript is potentially interesting for the journal provided the authors are willing to address some minor points raised by the reviewers.

We would appreciate receiving your revised manuscript by Oct 21 2019 11:59PM. To enhance the reproducibility of your results, we recommend that if applicable you deposit your laboratory protocols in protocols.io, where a protocol can be assigned its own identifier (DOI) such that it can be cited independently in the future. For instructions see: http://journals.plos.org/plosone/s/submission-guidelines#loc-laboratory-protocols

We look forward to receiving your revised manuscript.

Kind regards,

Prof. Raffaele Serra, M.D., Ph.D

Academic Editor

PLOS ONE

Journal Requirements:

Additional Editor Comments (if provided):

The manuscript need some revisions. See reviewers' comments.

Reviewers' comments:

Reviewer's Responses to Questions

**Comments to the Author**

1. Is the manuscript technically sound, and do the data support the conclusions?

Reviewer #1: Yes

2. Has the statistical analysis been performed appropriately and rigorously? 

Reviewer #1: Yes

3. Have the authors made all data underlying the findings in their manuscript fully available?

Reviewer #1: Yes

4. Is the manuscript presented in an intelligible fashion and written in standard English?

Reviewer #1: Yes

5. Review Comments to the Author

Reviewer #1: General Comments:

The authors present a paper discussing the presence or absence of pulmonary vascular abnormalities in patients with hepatopulmonary syndrome (HPS). The authors evaluated the lung CT scans of a cohort of HPS patients to determine the presence of abnormalities that could explain HPS, but found no correlation between vascular changes in the lung and presence of HPS. The findings from this study are a valuable and novel contribution to the literature. Minor changes are suggested below.

Title: The title is a bit misleading as it seems to suggest a relationship between HPS and pulmonary vascular abnormalities. The title could be re-worded to more accurately reflect the findings. For example “Prevalence of pulmonary vascular abnormalities on chest computed tomography do not relate to hepatopulmonary syndrome”.

Abstract: Abstract is well written and appropriately reflects the remainder of the submission. Some changes are required. In the second sentence of the abstract – please change “access” to “assess”. All locations that have O2 in the abstract and main text should have the 2 as a subscript. Currently, they are all superscript. Finally, the first sentence of the conclusion should state “The prevalence of pulmonary…”

Introduction: Concise and well written. No significant changes recommended.

Materials and Methods: The first paragraph under “Patients” needs some re-wording. The second sentence in the first paragraph in particular seems incomplete and reads awkwardly. It appears the authors are trying to state how the patients were selected, but the sentence is a fragment. In addition, the first criteria (i) seems too vague. Presence of liver disease is a wide ranging statement and could represent acute or chronic. The authors should be more specific here (e.g. chronic liver disease with portal hypertension). The remainder of the methodology and statistical analysis description appear appropriate.

Results: Results are presented appropriately. A clarification is required, though. In the paragraph that begins “There was no difference in the prevalence of pulmonary vascular abnormalities when comparing normoxemic (n=20) and hypoxemic patients (n=18) (p=0.513),” the final sentence is incomplete. Presumably, the authors were trying to state there was no correlation between PaO2 and ABR; however, please adjust accordingly.

Discussion: The discussion appears thorough and well written. The third to last sentence is written as “The prevalence of these abnormalities on Chest CT is low thanand

were not correlated with PaO²,” and requires spelling and grammatical correction.

Figures, Figure Legends: Appropriate

Tables: Appropriate.

References: Appropriate.

Summary: Overall, the paper is well written and addresses an area regarding HPS that requires clarification. However, the submission does require minor changes, mostly grammatical, detailed above.

6. PLOS authors have the option to publish the peer review history of their article (what does this mean?). If published, this will include your full peer review and any attached files.

Reviewer #1: No

---

## [Author Response · Author response to Decision Letter 0]

24 Sep 2019

September 20th 2019

Prof. Raffaele Serra, M.D., Ph.D

Academic Editor

PLOS ONE

Dear Editor:

 I wish to submit an revised manuscript for publication in the PLOS ONE , titled “Hepatopulmonary syndrome has a low prevalence pulmonary vascular abnormalities on chest computed tomography.” Felipe S.Torres, Juliana F. Zampieri, Betina C. Machado, Marli M. Knorst and Marcelo B. Gazzana coauthored the paper. Bellow the modifications required.

Please ensure that your manuscript meets PLOS ONE's style requirements, including those for file naming.

 The manuscript was reviewed to meet PLOS ONES style requirements.

Reviewer #1: The title is a bit misleading as it seems to suggest a relationship between HPS and pulmonary vascular abnormalities. The title could be re-worded to more accurately reflect the findings.

 Thank you for this comment. The title was modified to “Low prevalence of pulmonary vascular abnormalities on chest computed tomography in patients with hepatopulmonary syndrome.”

Reviewer #1: Abstract is well written and appropriately reflects the remainder of the submission. Some changes are required. In the second sentence of the abstract – please change “access” to “assess”. All locations that have O2 in the abstract and main text should have the 2 as a subscript. Currently, they are all superscript. Finally, the first sentence of the conclusion should state “The prevalence of pulmonary…”

 Thank you for all observations. All suggestions were modified accordingly and the abstract was appropriately corrected, as follows:

 “This study aimed to assess the prevalence of type 1 and 2 pulmonary vascular abnormalities on chest computed tomography (CT)”

 “The prevalence of pulmonary vascular abnormalities on chest CT of patients with cirrhosis and HPS is low and not correlated with PaO2.”

Reviewer #1: The first paragraph under “Patients” needs some re-wording. The second sentence in the first paragraph in particular seems incomplete and reads awkwardly. It appears the authors are trying to state how the patients were selected, but the sentence is a fragment. In addition, the first criteria (i) seems too vague. Presence of liver disease is a wide ranging statement and could represent acute or chronic. The authors should be more specific here (e.g. chronic liver disease with portal hypertension).

 “Patients were included where they had a chest CT and a diagnosis of HPS based on the following criteria: (i) presence of chronic liver disease; (ii) alveolar–arterial oxygen gradient (AaO2) >15 mmHg (20 mmHg in patients over 64 years old) detected with blood gas analysis; and (iii) demonstration of intrapulmonary vascular dilatation by means of a positive contrast-enhanced echocardiography or perfusion lung scanning with technetium-99m-labelled macroaggregated albumin (1,2.)”.

Reviewer #1: A clarification is required, though. In the paragraph that begins “There was no difference in the prevalence of pulmonary vascular abnormalities when comparing normoxemic (n=20) and hypoxemic patients (n=18) (p=0.513),” the final sentence is incomplete. Presumably, the authors were trying to state there was no correlation between PaO2 and ABR; however, please adjust accordingly.

 “There was no difference in the prevalence of pulmonary vascular abnormalities when comparing normoxemic (n=20) and hypoxemic patients (n=18) (p=0.513). The mean PaO2 of patients with any vascular pulmonary abnormality on CT (n=11) was 82.1 mmHg ± 17.9 and for those without any pulmonary vascular abnormality (n=27) it was 77.9 mmHg ± 15.8 (p=0.515). There was no correlation between PaO2 and ABR (Figure 3).”

Reviewer #1: Discussion: The discussion appears thorough and well written. The third to last sentence is written as “The prevalence of these abnormalities on Chest CT is low thanand were not correlated with PaO²,” and requires spelling and grammatical correction.

“The prevalence of these abnormalities on Chest CT were not correlated with PaO2.”

---

## [Editor Report · Decision Letter 1]

30 Sep 2019

Hepatopulmonary syndrome has  low prevalence of pulmonary vascular abnormalities on chest computed tomography

PONE-D-19-21631R1

Dear Dr. Folador,

We are pleased to inform you that your manuscript has been judged scientifically suitable for publication and will be formally accepted for publication once it complies with all outstanding technical requirements.

With kind regards,

Prof. Raffaele Serra, M.D., Ph.D

Academic Editor

PLOS ONE

Additional Editor Comments (optional):

amended manuscript is acceptable
---

## [Editor Report · Acceptance letter]

9 Oct 2019

PONE-D-19-21631R1 

Hepatopulmonary syndrome has low prevalence of pulmonary vascular abnormalities on chest computed tomography 

Dear Dr. Folador:

I am pleased to inform you that your manuscript has been deemed suitable for publication in PLOS ONE. Congratulations! Your manuscript is now with our production department. 

With kind regards,

on behalf of

Prof. Raffaele Serra 

Academic Editor

PLOS ONE